# Impaired Remodeling of White Adipose Tissue in Obesity and Aging: From Defective Adipogenesis to Adipose Organ Dysfunction

**DOI:** 10.3390/cells13090763

**Published:** 2024-04-30

**Authors:** Carla Iacobini, Martina Vitale, Jonida Haxhi, Stefano Menini, Giuseppe Pugliese

**Affiliations:** Department of Clinical and Molecular Medicine, “La Sapienza” University, 00189 Rome, Italy; carla.iacobini@uniroma1.it (C.I.); martina.vitale@uniroma1.it (M.V.); jonida.haxhi@uniroma1.it (J.H.); stefano.menini@uniroma1.it (S.M.)

**Keywords:** white adipose tissue, remodeling, adipogenesis, obesity, cardiometabolic risk, aging, age-related conditions

## Abstract

The adipose organ adapts and responds to internal and environmental stimuli by remodeling both its cellular and extracellular components. Under conditions of energy surplus, the subcutaneous white adipose tissue (WAT) is capable of expanding through the enlargement of existing adipocytes (hypertrophy), followed by de novo adipogenesis (hyperplasia), which is impaired in hypertrophic obesity. However, an impaired hyperplastic response may result from various defects in adipogenesis, leading to different WAT features and metabolic consequences, as discussed here by reviewing the results of the studies in animal models with either overexpression or knockdown of the main molecular regulators of the two steps of the adipogenesis process. Moreover, impaired WAT remodeling with aging has been associated with various age-related conditions and reduced lifespan expectancy. Here, we delve into the latest advancements in comprehending the molecular and cellular processes underlying age-related changes in WAT function, their involvement in common aging pathologies, and their potential as therapeutic targets to influence both the health of elderly people and longevity. Overall, this review aims to encourage research on the mechanisms of WAT maladaptation common to conditions of both excessive and insufficient fat tissue. The goal is to devise adipocyte-targeted therapies that are effective against both obesity- and age-related disorders.

## 1. Introduction

Adipose tissue (AT) is a fascinating and multifaceted organ. From a morphological and spatial–topographic perspective, it can delineate the contour of a physically fit individual, accentuating the elegant curves and captivating lines with artistic precision. Conversely, in the case of a long-lasting unhealthy lifestyle, particularly when coupled with predisposing genetic factors, AT can become a burdensome and disabling factor, leading to physical and psychological distress. Alongside the physical challenges associated with obesity, such as severely restricted mobility and diminished quality of life, exists the psychological burden of perceiving AT as a catalyst for social judgment and isolation.

The eclectic nature of this large organ is also evident from a functional perspective. While it was traditionally perceived as an inert depot for storing triacylglycerol, a mounting body of evidence over the past 50 years indicates that AT functions as a dynamically active metabolic and endocrine organ [1]. Its functions range from providing mechanical and thermal insulation to protect other organs to playing a critical role in energy storage, heat production, and immune function. To accomplish these functions, adipocytes are organized in specific areas known as white AT (WAT) and brown adipose tissue (BAT), with further divisions based on their location in the body. As we delve deeper into the origins and functions of adipocytes, including their metabolic, bioenergetic, and immune functions, it becomes evident that these cells play diverse roles even within the same AT depot [2]. Furthermore, the AT is not solely composed of adipocytes; rather, it comprises the extracellular matrix and various cell types, including adipocyte precursor cells, preadipocytes, endothelial cells, vascular precursor cells, and immune cells [3], collectively referred to as the stromal vascular fraction (SVF), also known as the AT microenvironment [4]. Advanced techniques like single-cell RNA sequencing (scRNA-seq) have offered deeper insights into WAT complexity, but they also raised new questions about the functions of its cellular subtypes [5,6].

The AT is able to adapt and respond to internal and environmental stimuli by remodeling both its cellular and extracellular components, which makes it a key player in health and disease [7]. This ability to undergo phenotypic, metabolic, and structural adaptations is called AT plasticity [8]. Dynamic AT changes in response to cold and warm temperatures involve the processes of beiging/browning and whitening, respectively, of WAT and BAT, the corresponding switch from a “storage and release” to an “uptake and oxidation” metabolic program and vice versa, and de novo adipogenesis [8]. This issue exceeds this review’s scope, and readers are referred to the vast literature on BAT plasticity and its role in thermogenesis.

Focusing on WAT plasticity, it is referred to as the ability of this fat depot to undergo adaptations to several stimuli, including food availability and intake, which trigger differentiation from precursors to mature adipocytes (i.e., adipogenesis), switching from a “fat storage” to a “fat mobilization” metabolic program, and WAT expansion through adipocyte hypertrophy and hyperplasia and vice versa, but also morphological and functional changes in various stromal vascular components [8]. These adaptations are essential for maintaining the health and function of WAT as both an energy storage and an endocrine organ. In addition, they have profound implications for modulating whole-body metabolism by regulating other important metabolic tissues, such as the liver, muscle, and pancreas, and for maintaining the health of cardiovascular, renal, nervous, and osteoarticular tissues.

The accumulation of visceral WAT (vWAT) around and within internal organs is linked to metabolic derangements, like insulin resistance and type 2 diabetes (T2D), which contrasts with subcutaneous WAT (sWAT), which is often viewed as protective, although this view is simplistic [2]. Furthermore, the aging process influences WAT through alterations in distribution patterns and composition, concomitant with diminished functionality of adipocyte precursor cells and an increase in the presence of senescent cells [9].

The focus of this review is the role of maladaptive WAT remodeling in obesity and associated comorbidities, with a special emphasis on impaired adipogenesis. In particular, we will discuss how WAT maladaptation may result from various defects in adipogenesis, leading to different WAT features and metabolic consequences. Additionally, we offer insights into the latest advancements in comprehending the molecular and cellular processes underlying age-related WAT maladaptation, highlighting its importance as a core element in aging and related chronic disorders, such as osteoarthritis, osteoporosis, cognitive dysfunction, and cachexia, and examining its potential as a therapeutic target to impact both elderly well-being and lifespan. By identifying significant unresolved questions, our objective is to highlight the hurdles that remain in envisioning and developing novel therapeutic interventions to target maladaptive adipocytes in obesity and age-related diseases.

## 2. Impaired White Adipose Tissue Remodeling in Obesity

The sWAT is the largest and safest WAT depot of the body, as it is capable of expanding under conditions of energy surplus through de novo adipogenesis, i.e., through recruitment of new adipocytes (hyperplasia), in addition to enlargement of existing adipocytes (hypertrophy), thus allowing accommodation of large amounts of lipids without compromising metabolic health [10]. Adipocyte size was shown to increase with expanding body fat up to a maximum, after which a more rapid increase in adipocyte numbers occurred [11]. Indeed, although the number of adipocytes in human sWAT is established around puberty, there is an annual turnover of ~10% [12].

However, once this adipogenic capacity is overwhelmed, sWAT expansion proceeds exclusively through hypertrophy, and when adipocytes achieve a critical cell volume, they degenerate and die through pyroptosis [13], with the formation of crown-like structures of dead/dying adipocytes surrounded by infiltrating macrophages and fibrotic tissue [14]. This results in sWAT dysfunction (adiposopathy) with fat diversion to vWAT and ectopic sites, including the liver, muscle, pancreas, and heart, ultimately leading to tissue and systemic inflammation and insulin resistance, development of T2D, and increased risk of cardiovascular disease (CVD) and cancer [15,16]. Thus, at variance with hyperplastic obesity, hypertrophic obesity is associated with increased cardiometabolic risk (Figure 1).

It is worth noting that the adipogenic capacity of sWAT is not only limited but also extremely variable among individuals, as shown by the existence of persons with obesity either with or without insulin resistance and components of the metabolic syndrome, including impaired glucose regulation, atherogenic dyslipidemia, and hypertension, despite the same body mass index (BMI). In addition to these two obese phenotypes, which are called metabolic unhealthy obese (MUO) and metabolic healthy obese (MHO), respectively [17,18], there are also lean phenotypes characterized by the presence or absence of metabolic derangement, i.e., the metabolic unhealthy normal weight (MUNW), also called metabolically obese normal weight [MONW], and the metabolic healthy normal weight (MHNW), respectively [19]. This indicates that it is not just a matter of absolute quantity of fat and shifts the focus on sWAT dysfunction and remodeling of fat depots, which is dependent on the individual capacity to expand and accommodate the surplus of energy [20]. However, it is debated whether MHO is a transient condition that is destined, sooner or later, to shift towards MUO [18], and studies in people with the MHO phenotype have shown that they still have an increased risk of T2D and CVD [21,22]. Nonetheless, there is no doubt that these individuals have a higher adipogenic capacity than those with the MUO and especially the MUNW phenotypes due to genetic and epigenetic factors [10].

### 2.1. Adipocyte Size as a Marker of White Adipose Tissue Dysfunction

The concept of impaired sWAT adipogenesis and expandability, linking obesity to its complications [23], points to the importance of adipocyte size (hypertrophy) as opposed to adipocyte numbers (hyperplasia) as a marker of sWAT dysfunction and predictor of increased cardiometabolic risk [24]. Adipocyte size varies according to several factors, including sex and fat depot (lower in vWAT than in sWAT in women, but not in men) [24], as well as the method of measurement [25]. Adipocytes are generally classified by size into three categories, i.e., small (<70 μm), large (70–120 μm), and very large (>120 μm) [25], or four categories, i.e., small, medium, large, and very large [26]. In this section, we will briefly summarize the evidence supporting the concept that the higher the size (or the lower the number) of sWAT adipocytes, the higher the risk of fat diversion to vWAT and ectopic sites and the impairment of insulin sensitivity and glucose homeostasis predisposing to CVD and cancer. In particular, we will focus on the studies showing that metabolic dysfunction correlates with “large adipocytes”, indicating the predominance of hypertrophy over hyperplasia, but also on the reports highlighting the importance of a population of “small adipocytes” in the development of insulin resistance and T2D, possibly suggesting impaired adipocyte maturation.

#### 2.1.1. Large Adipocytes

Adipocyte size was shown to increase with increasing body fat mass, with a curvilinear relationship [24]. Likewise, adipocyte size was found to increase with increasing BMI until reaching a plateau at a BMI ~30 kg/m^2^, which, however, was lower in men (~25 BMI kg/m^2^) than in women (~35 kg/m^2^), suggesting a greater propensity to develop hypertrophic obesity in the male sex compared to the female sex [24]. Several studies have shown that adipocyte size is increased in people with obesity [11,27,28] and T2D [29,30,31], and even in lean individuals with a genetic predisposition to T2D [32,33] but not to obesity [32]. Moreover, adipocyte size correlated positively with lipolysis, WAT, systemic inflammation, and insulin resistance [27,29,30,31,34] and negatively with lipogenesis [34], and it predicted the incidence of T2D [30,31] as well as the persistence of T2D or T2D risk after gastric by-pass surgery [35]. A number of studies have reported an association between adipocyte size and dyslipidemia, with differences according to sex and fat depot [36,37,38,39,40]. Conversely, adipocyte numbers correlated positively with insulin sensitivity and HDL-cholesterol and negatively with insulin and triglycerides [36], and a decreased adipogenic potential of sWAT, but not vWAT, was found to be associated with visceral obesity [41] and impaired glucose regulation [42].

When using the deviation of the measured volume from the expected volume according to body fat mass, with positive and negative values corresponding to hypertrophy and hyperplasia, respectively, this marker of adipocyte morphology was independent of sex and body weight and correlated positively with fasting plasma insulin levels and insulin resistance, irrespective of adipocyte size, and negatively with total adipocyte numbers [41]. Moreover, a negative correlation was observed between adipocyte hypertrophy and hyperplasia when adjusted for the predicted adipocyte volume at a given body fat mass, indicating that the higher the predicted adipocyte size, the lower the rate of adipogenesis. In fact, the absolute number of new adipocytes generated each year was 70% lower in hypertrophy than in hyperplasia, and individual values correlated strongly with morphology but not adipocyte size [43].

However, other studies did not find a correlation between sWAT adipocyte size and metabolic derangement [28,44,45] or reported an association in non-diabetic but not in T2D individuals [46]. Furthermore, in Asian Indians, size was higher but less strongly correlated with measures of adiposity and metabolic parameters in sWAT versus vVAT adipocytes [47].

#### 2.1.2. Small Adipocytes

A number of studies have shown a bimodal cell size distribution in WAT, with about the same proportion of large and small adipocytes and the small rather than the large cell population being related to metabolic derangement associated with obesity. In fact, McLaughlin et al. reported that among individuals with obesity, those with insulin resistance had fewer large adipocytes with a similar or larger mean diameter and a higher ratio of small to large cells compared with their insulin-sensitive counterparts. It is worth noting that the fraction of small adipocytes was associated with insulin resistance, decreased expression of adipogenic genes, and increased expression of inflammatory genes, whereas the diameter of the large adipocytes was associated with percent body fat [48,49,50]. Moreover, Pasarica et al. showed that obese individuals with T2D had lower total adipocyte numbers, with an increase in the fraction of small adipocytes as well as a decrease in the size of small and medium adipocytes and a marked increase in the size of very large adipocytes compared to those without T2D [51]. Likewise, Fang et al. found an increase in the percentage of small cells and a corresponding decrease in the percentage of large cells in both sWAT and vWAT of individuals with versus without T2D; the fraction of small adipocytes, but not that of large adipocytes, correlated positively with indices of glucose homeostasis, though this association was significant only for vWAT [52]. Finally, two studies assessed the impact of short-term overfeeding on metabolic derangement according to adipocyte size. Johannsen et al. showed no association between sWAT cell size and ectopic fat accumulation, but individuals with smaller adipocytes at baseline had a greater decrease in insulin sensitivity, which was related to skeletal muscle tissue inflammation [53]. McLaughlin et al. reported that insulin-resistant individuals, who had a higher adipocyte diameter and percentage of small adipocytes associated with a worse metabolic profile, exhibited no change in cell size and no (or only minor) further fat diversion to vWAT and ectopic sites and deterioration of insulin sensitivity with diet-induced body weight gain. This was at variance with insulin-sensitive individuals, who demonstrated significant enlargement of adipocytes and reduction of the percentage of small adipocytes associated with a marked impairment of insulin sensitivity, which was predicted by the smaller baseline adipocyte size [54].

These findings were thought to challenge the AT expandability hypothesis linking adipocyte hypertrophy to metabolic derangement associated with obesity [55]. However, they have also been interpreted as the consequence of an impaired ability of sWAT to generate fully mature adipocytes capable of accommodating surplus energy [10]. This implies that sWAT dysfunction limiting tissue plasticity and fat storage capacity may arise not only from a reduced hyperplastic WAT response due to impaired recruitment and expansion of adipocyte precursors (the first step of adipogenesis) but also from a failure of adipocytes to accumulate lipids due to defective terminal differentiation of preadipocytes into fully mature cells (the second step of adipogenesis) [56].

### 2.2. Adipogenesis as a Central Mechanism for White Adipose Tissue Plasticity

As discussed above, adipogenesis represents a safe mechanism for WAT expansion under conditions of energy surplus, as it allows for accommodating large amounts of fat without achieving the critical adipocyte volume that drives cell degeneration and death as well as fat diversion to ectopic sites and related metabolic derangement [8].

Adipogenesis is the process by which pluripotent mesenchymal stem cells (MSCs) commit to the adipose lineage to become preadipocytes, which then differentiate into adipocytes [57]. Preadipocytes differentiate by undergoing mitotic clonal expansion, followed by growth arrest and terminal differentiation with acquisition of a mature phenotype, i.e., expression of adipocyte genes and accumulation of triglycerides (Figure 2) [57].

Studies of pluripotent stem cells or preadipocytes, such as the C3H10T1/2 and 3T3-L1/F442A cell lines, respectively, have enabled the characterization of the two steps of adipogenesis and the identification of the molecular regulators of this process. The results of these studies have been extensively reviewed elsewhere [57,58] and will be briefly summarized here.

#### 2.2.1. Commitment of Pluripotent Stem Cells to Adipocyte Lineage

Pluripotent MSCs, which are located in the SVF of AT and the bone marrow, can undergo a multistep process of commitment in the adipocyte, myocyte, chondrocyte, or osteocyte lineage, depending on specific signals. Commitment to the adipocyte lineage with the generation of preadipocytes is triggered by chronic overfeeding through increased GLUT4-dependsent glucose uptake in the AT [59]. Several factors have been identified that modulate commitment to the adipocyte lineage, including the members of the bone morphogenetic protein (BMP) family BMP-2 and 4 [60,61], the wingless-type MMTV integration site (Wnt) family [62,63,64], and the Hedgehog (Hh) protein family [65]. Interestingly, these factors are members of developmental signaling pathways that have been conserved from invertebrates to vertebrates, like fat function and distribution [66].

The BMPs play a fundamental role in the commitment to adipocyte lineage by signaling through the BMP receptors (BMPr) 2 and 1, which form cell surface complexes with serine/threonine kinase activity. Upon binding to BMPs, the BMPr1 kinase is phosphorylated and activated, thus phosphorylating the transcription factor Smad 1/5/8, which forms a complex with Smad4 that translocates into the nucleus and regulates gene expression [61]. The non-canonical BMP signaling through the mitogen-activated protein kinases (MAPK) extracellular signal-regulated kinase (ERK), Jun N-terminal kinase (JNK), and p38 is also involved in the promotion of adipogenesis [67]. Downstream targets of BMP signaling include three cytoskeleton-associated proteins, i.e., lysyl oxidase (Lox), translationally controlled tumor protein 1 (Tpt1), and αβ-crystallin, which promote F-actin fibers’ breakdown and rearrangement to the preadipocyte periphery, resulting in cell shape changes [68]. Signaling through the BMP pathway is inhibited by several substances, including gremlin, noggin, matrix Gla protein (MGP), BMP, and activin membrane bound inhibitor (BAMBI), chordin-like, follistatin, and follistatin-like 3 [67].

The Wnt family of secreted signaling glycoproteins also participates in stimulating adipocyte lineage commitment via the canonical pathway by acting upstream of BMPs. The activation of Wnt signaling through the binding of Wnt ligands to the cell surface frizzled receptor and the associated low-density lipoprotein receptor-related coactivator 5/6 results in the dissociation of the destruction complex consisting of adenomatous polyposis coli (APC), axin, and glycogen synthase kinase-3β (GSK-3β). This prevents phosphorylation by GSK-3β and consequent ubiquitination and proteasomal degradation of β-catenin, which allows it to translocate into the nucleus and activate the transcription factors lymphoid enhancer factor 1 (Lef1) and T-cell factor (Tcf) [69]. Using microarray technology, the activators R-spondins-2 and -3 and the downstream transcription factors Lef1 and Tcf of the canonical Wnt signaling pathway were found to be upregulated in proliferating A33 preadipocytes but not in C3H10T1/2 progenitor cells, together with β-catenin nuclear accumulation [62]. Though these findings might suggest a favorable effect of the Wnt pathway on adipocyte commitment, the latter was shown to be inhibited by the Wnt inducible signaling pathway protein (WISP2) via negative regulation of the BMP pathway (see below) [70]. Moreover, the activation of the Wnt pathway plays an inhibitory role in adipogenesis by maintaining preadipocytes in undifferentiated states by suppressing PPARγ and C/EBPα, thus promoting diversion of precursor cells to other lineages [53,64].

Finally, it is generally agreed that expression of Hh proteins inhibits adipocyte lineage commitment. These proteins bind to a cell surface receptor complex consisting of the Patched (Ptc) receptors 1 and 2 and the seven-transmembrane receptor Smoothened (Smo), thus relieving Smo from Ptc inhibition and allowing for the activation of the transcription factors Gli1, Gli2, and Gli3 [71]. In C3H10T1/2 progenitor cells, sonic Hh protein was shown to enhance BMP-2-stimulated osteoblast commitment while suppressing adipocyte commitment [65].

#### 2.2.2. Preadipocyte Differentiation into Mature Adipocytes

Mitotic clonal expansion is a prerequisite for preadipocyte differentiation [72]. In vitro, when exposed to a differentiation medium containing insulin, dexamethasone, and isobutyl-methyl-xanthine, a non-selective phosphodiesterase inhibitor raising intracellular cyclic AMP, preadipocyte cell lines, such as 3T3-L1 and 3T3-F442A, reenter the cell cycle and initiate a proliferative burst. Cell cycle progression is mediated by cyclin-dependent kinases (CDKs) that phosphorylate the retinoblastoma cell-cycle-related protein (Rb), thus releasing E2 transcription factor (E2F) 1, which becomes available to promote the expression of genes involved in S-phase entry, DNA synthesis, and mitosis [73]. The protein kinase A-dependent phosphorylation of cyclic AMP response element-binding protein (CREB) is a critical step in this process, as phosphorylated CREB binds to the promoter and increases the expression of cyclin D1 [74] and the transcription factor CCATT enhancer binding protein (C/EBP) β [75]. While cyclin D1 is required for proliferation [74], C/EBPβ undergoes dual phosphorylation sequentially induced by MAPK and GSK-3β [76], which produces a conformational change that renders the leucine zipper of monomeric C/EBPβ accessible for dimerization, thus allowing acquisition of DNA-binding activity [77]. Though C/EBPβ is required for mitotic clonal expansion in vitro [78], concomitant expression and activation of C/EBPδ also play a role in this process [79]. Preadipocyte factor 1 (Pref-1)/delta-like non-canonical notch ligand 1 (Dlk1) is a transmembrane protein expressed by preadipocytes that inhibits mitotic clonal expansion and hence differentiation into mature adipocytes [80].

Induction of terminal differentiation requires termination of mitotic clonal expansion by exiting the cell cycle, and it is mediated by C/EBPβ-induced transactivation of C/EBPα and peroxisome proliferator-activated receptor (PPAR) γ [81]. The genes coding for these transcription factors have C/EBP regulatory elements for C/EBPβ binding [82], through which C/EBPα and PPARγ cross-activate each other [83] and C/EBPα maintains its expression (autoactivation) [84]. In vitro adipocyte differentiation requires PPARγ [85], which acts as the master regulator of the adipogenesis program through a unified pathway [86]. It was in fact shown to promote adipogenesis in C/EBPα-deficient cells [73], which is at variance with C/EBPα in cells lacking PPARγ [87]. However, repression of E2F1-dependent transcription by C/EBPα was found to be critical for the suppression of cell proliferation and the induction of terminal differentiation [88]. Transactivated C/EBPα and PPARγ serve as pleiotropic transcriptional activators of a group of genes promoting the acquisition of the morphological and functional features of mature adipocytes [89]. Among these genes, the transcription factor sterol regulatory element-binding protein (SREBP) 1c, primarily regulated at the transcriptional level by insulin, plays a central role by controlling the expression of genes coding for enzymes of fatty acid (FA) and triacylglycerol biosynthesis and catabolism, such as acetyl-CoA carboxylase (ACC), fatty acid synthase (FAS), and stearoyl CoA desaturase (SCD1) [90]. In addition, it activates PPARy expression through the production of an endogenous ligand [91]. For this reason, it is also called adipocyte determination and differentiation factor-1 (ADD-1). As a result of de novo lipogenesis, preadipocytes lose their fibroblast-like appearance and acquire that of mature adipocytes by accumulating cytoplasmic triglycerides. Other genes that are induced during adipocyte differentiation include those coding for the lipolytic enzymes adipocyte-triacylglycerol lipase (ATGL) and hormone-sensitive lipase (HSL) [92], the gluconeogenic enzyme phosphoenolpyruvate carboxykinase (PEPCK), which provides glycerol 3-phosphate for FA re-esterification [93], the carrier protein fatty acid binding protein (FABP) 4 or 422/adipocyte protein 2 (aP2), which is involved in FA uptake, transport, and metabolism [94], the insulin receptor and the insulin-responsive glucose transporter GLUT4, and the adipokines leptin, resistin, and adiponectin [57].

Several factors were shown to regulate adipogenesis by stimulating or inhibiting terminal differentiation via modulation of the adipogenic transcription factors PPARγ and C/EBPα. Activation of BMP signaling has a stimulatory role, as phosphorylated Smad 1/5/8 releases zinc finger protein (Zfp) 423 from the complex with WISP2, thus enabling Zfp423 to translocate into the nucleus and activate PPARγ transcription [70]. Moreover, the transcription factor Schnurri2 was also found to mediate BMP-dependent induction of PPARγ expression [65]. The Wnt signaling pathways was shown to exert an inhibitory effect on the differentiation of preadipocytes into mature adipocytes not only via formation of the WISP2-Zpf423 complex but also through the stabilization and nuclear translocation of β-catenin [63,64]. Likewise, an inhibitory role in adipocyte differentiation was also reported for the Hh signaling pathway, which is mediated by GATA 2and 3 [66]. At variance with E2F1, E2F4 was found to negatively regulate adipogenesis by repressing PPARγ expression during terminal adipocyte differentiation [95]. Hypophosphorylated RB proteins are also involved by playing a dual role, as they were shown to either stimulate adipogenesis by enhancing the binding and transcription of C/EBPβ [96] or inhibit adipogenesis by forming a complex with PPARγ, which contains the histone deacetylase (HDAC) 3, which attenuates PPARγ-induced expression of adipogenic genes [97]. The Krupel-like factors (KLFs) are deoxyribonucleic acid-binding transcriptional factors that regulate adipogenesis both positively and negatively [98]. Activators of adipogenesis include KLF4, early growth response 2 (Egr2)/Krox20, KLF5, KLF6, KLF8, KLF9, KLF13, and KLF15, which act via C/EBPs and PPARγ, except KLF6, which inhibits the expression of Pref-1/Dlk1 [99]. Suppressors of adipogenesis include KLF2, KLF3, and KLF7, which also act through C/EBPs and PPARγ [99]. Finally, galectins are multifunctional lectins belonging to an ancient family defined by recognition of β-galactoside structures through a carbohydrate-recognition-binding domain (CRD). Three of the mammalian galectins, galectin-1 [100], 3 [101,102], and 12 [103], have been shown to be required for in vitro adipocyte differentiation. Moreover, it was found that (a) galectin-3 stimulates proliferation of preadipocytes [104]; (b) both galectin-1 and 3 interact with PPARγ during differentiation of preadipocytes [100,101]; and (c) galectin-12 causes cell cycle arrest when expressed in cultured cells [105], negatively regulates lipolysis in adipocytes [106], and promotes apoptosis in WAT [107].

### 2.3. Defective Adipogenesis and White Adipose Tissue Dysfunction

The classical view linking impaired sWAT expandability due to defective adipogenesis to tissue and systemic inflammation and associated metabolic derangement has been tested in in vivo studies using genetically modified mice maintained under standard conditions or challenged with energy surplus by feeding a high-fat diet (HFD) or crossing with mice with genetic obesity. These animals were either lacking or overexpressing one of the numerous factors that were shown to be involved in the formation of mature adipocytes from uncommitted precursors through the two steps of adipogenesis in in vitro studies. In some cases, the genetic modification was responsible for a phenotype resembling human lipodystrophy, which allowed for investigating the effect of reduced AT development on the occurrence of metabolic derangement and the response to overfeeding. In some others, it resulted in a silent phenotype that required exposure to increased energy/fat intake to unmask the impaired WAT capacity to safely accommodate lipids. However, studies in animal models have been difficult for several reasons. The most important one is that ablation of genes coding for some of the factors involved in adipogenesis is lethal, and alternative strategies had to be used. One of these strategies was generating adipocyte-specific knockout mice using the Cre-loxP system, which, however, showed different specificity and efficacy depending on the gene promoter. Here, we describe the results of a number of these studies investigating the role of various molecular regulators, which might help to understand the mechanisms operating in the human obese and lean phenotypes (Table 1).

#### 2.3.1. Conserved Developmental Signaling Pathways

Mice overexpressing BMP4 in adipocytes from the FABP4/aP2 promoter showed reduced sWAT mass and adipocyte size along with an increased number of white adipocytes with brown characteristics, increased energy expenditure, improved insulin sensitivity and glucose tolerance, and protection against HFD-induced obesity and metabolic derangement [108]. Conversely, adipocyte-specific BMP4-deficient mice exhibited increased WAT mass and adipocyte size along with brown adipocytes with white characteristics and impaired insulin sensitivity and glucose tolerance [108]. Consistent with these results, ablation of the BMP enhancer kielin/chordin-like protein aggravated HFD-induced obesity, whereas its overexpression was protective [109], and deletion of the BMP inhibitor follistatin-like 3 was shown to reduce body fat and improve insulin sensitivity and glucose tolerance [110]. However, combined loss of follistatin and follistatin-like 3 was associated with increased fat mass, insulin resistance, and glucose intolerance [110].

Mice overexpressing the Wnt ligand Wnt10b in adipocytes from the FABP4/aP2 promoter were shown to have impaired WAT (and BAT) development, with a reduction in the number but not in the size of adipocytes [111] and protection from dietary and genetic obesity [111,112]. The finding that these mice were more insulin-sensitive and glucose-tolerant than their wild-type counterparts despite the impaired WAT plasticity was attributed to decreased levels of resistin and/or inflammatory cytokines [111,112]. Resistance to HFD-induced obesity was also observed in mice with ablation of β-catenin [113] or the chaperone protein Wntless [114] in mature adipocytes as well as in mice with global deficiency of functional secreted frizzled-related protein (SFRP) 5, an inhibitor of Wnt signaling [115]. Interestingly, WAT of mice lacking functional SFRP-5 had an unchanged number of adipocytes associated with an increase in small cells and a decrease in large cells [115]. Conversely, increased body weight, WAT mass, and adipocyte size associated with impaired insulin sensitivity and glucose tolerance were observed in HFD-fed mice with adipocyte-specific deletion of transcription factor 7-like 2 (TCF7L2), a key intracellular effector of the Wnt-signaling pathway [116].

The naturally occurring spontaneous mesenchymal dysplasia (mes) in adult mice (*Ptc1^mes/mes^*) carrying a deletion of Ptc1 at the carboxyl-terminal cytoplasmic region, which is essential for the inhibition of Hh signaling, showed a significant reduction in epididymal WAT mass and adipocyte size (and possibly number) under standard feeding conditions. In addition, the gene expressions of PPARγ and CEBPα and adipocyte markers were reduced, with no change in glucose tolerance [117]. However, previous studies have provided opposite results, which contrast with the inhibitory role of the Hh proteins on adipocyte commitment and differentiation. In fact, body weight and fat mass were significantly increased by injection of the Hh agonist sonic Hh-IgG fusion protein in adult mice [118] and decreased by injection of anti-Hh antibodies in mice with dietary or genetic obesity [119].

#### 2.3.2. CCATT Enhancer Binding Proteins

Studies in animal models lacking C/EBPs were severely limited by the embryonic lethality of mice deficient in these transcription factors.

Mice knockout for C/EBPβ or C/EBPδ were shown to have virtually normal WAT mass, whereas a marked defect due to a reduction in the number more than in the size of adipocytes was found in double C/EBPβ and C/EBPδ knockout mice, suggesting a functional redundancy of members of the C/EBP family. However, it should be considered that only 15% of *cebpb*^−/−^*/cebpd*^−/−^ animals survived until adulthood [120].

As mice deficient in C/EBPα failed to accumulate lipids in sWAT but died shortly after birth from hypoglycemia due to impaired liver gluconeogenesis [121], transgenic *cebpa*^−/−^ animals expressing C/EBPα only in the liver were generated [122]. These mice also showed an absence of WAT associated with postprandial hyperlipidemia, fatty liver, and increased insulin but normal glucose levels [122].

#### 2.3.3. Peroxisome Proliferator-Activated Receptor γ

Ablation of PPARγ in vivo is also lethal; thus, alternative mouse models with either global or adipocyte-specific PPARγ deficiency have been used for dissecting the role of this transcription factor in adipogenesis [123].

Rescuing of embryonic lethality despite global PPARγ deletion was obtained by preserving PPARγ expression in the trophoblasts or generating hypomorphic, heterozygous, chimera, or isoform-specific knockout animals as well as mice carrying dominant negative human mutations [123]. Inactivation of PPARγ in the embryo but not in trophoblasts caused severe lipodystrophy, organomegaly, and insulin resistance, whereas glucose tolerance was decreased in male and increased in female mice [124]. Mice carrying a hypomorphic mutation at the PPARγ2 locus also showed no or only rare and hypertrophic adipocytes due to the elimination of PPARγ before WAT development, in addition to reduced premature lethality. However, surviving mice had no hepatic steatosis but rather fat accumulation at the muscle level, with only mild impairment of glucose tolerance and elevation of glucose and insulin levels only in the fed state, indicating modest insulin resistance. The discrepancy between the severe lipodystrophy resembling human autosomal dominant familial partial lipodystrophy and the mild metabolic phenotype was attributed to the induction of a compensatory gene expression program in the muscle promoting oxidation of excess lipids [125]. In contrast, mice heterozygous for PPARγ were shown to have comparable body weight, fat-pad mass, and free fatty acid (FFA) levels and to be more insulin-sensitive than their wild-type littermates, a finding that was explained by the lack of PPARγ-dependent transcriptional repression of genes impairing insulin sensitivity and/or increased expression of leptin [126]. The use of *pparg*^+/+^ and *pparg*^−/−^ chimeras demonstrated the central role of PPARγ in AT development, as only *pparg*^+/+^ adipocytes were detected in these animals [85]. Compared with wild-type animals, mice with deletion of PPARγ2 showed comparable body weight but reduced WAT mass due to a decrease in both adipocyte numbers and size, together with severe insulin resistance in male but not female animals and normal glucose tolerance. Moreover, these mice were protected from HFD-induced obesity [127]. In contrast, another study reported that *pparg*2^−/−^ mice fed standard chow exhibited no difference versus wild-type mice in body weight, WAT mass, adipocyte numbers and size, FFA and triglyceride levels, and ectopic fat accumulation, but they showed peripheral insulin resistance and impaired glucose tolerance [128]. Conversely, when fed an HFD, knockout mice showed increased WAT area and adipocyte size despite comparable body weight but similarly reduced insulin sensitivity [128]. These findings indicate that PPARγ2 regulates glucose homeostasis and insulin sensitivity independent of its effect on adipogenesis [127,128]. Heterozygous knock-in mouse models carrying the dominant negative PPARγ mutations L466A and P465L, which were found in humans with lipodystrophy and/or the metabolic syndrome, recapitulated some of the features of the human phenotypes. Heterozygous PPARγ L466A knock-in mice had lower body weight gain and fat mass with a heterogenous population of adipocytes of predominantly small and medium size instead of a homogenous population of large adipocytes, as in wild-type animals. This was associated with decreased adiponectin and increased FFA levels, hepatic steatosis, and hypertension but normal insulin sensitivity. However, on HFD, these mice gained less weight and developed a more severe insulin resistance than wild-type animals [129]. In contrast, heterozygous PPARγ P465L knock-in mice had normal body weight, fat mass, and insulin sensitivity but altered AT distribution with increased sWAT and a more homogeneous population of adipocytes of a larger size than in wild-type mice, associated with increased glucose tolerance and hypertension. When fed an HFD, these animals showed a comparable reduction in whole-body, hepatic, and peripheral insulin sensitivity compared to wild-type mice [130].

Mice with adipocyte-specific PPARγ deletion also showed a lipodystrophic phenotype, though the results of these studies were not univocal with respect to the extent of the reduction of fat mass and the metabolic consequences of impaired WAT development and/or maintenance. A less severe phenotype was observed in adipocyte-specific PPARγ knockout mice generated using a FABP4/aP2 gene-promoter-regulated Cre-loxP system, a finding that was attributed to the deletion of PPARγ at a later developmental stage. He et al. showed that these mice had reduced WAT (and BAT) mass due to a marked decrease in adipocyte numbers because of apoptosis, associated with macrophage infiltration and fibrosis [131]. It is worth noting that while most of the remaining cells were highly hypertrophic, there was also a population of small adipocyte-like cells, suggesting the recruitment of adipocyte precursors that failed to maturate, a feature that mimics that reported in obese humans [48,49,50,51]. Under standard chow feeding, these animals showed liver steatosis with decreased leptin and increased FFA, triglyceride levels, and hepatic glucose production compared with wild-type mice, though insulin sensitivity and glucose tolerance were comparable. However, when fed an HFD, they gained significantly less weight, and their residual WAT (and BAT) became more lipoatrophic as in aging mice on standard chow; moreover, glucose and insulin levels were two-fold higher and insulin sensitivity was lower than in wild-type animals. These findings highlight the essential role of PPARγ in the survival of mature adipocytes and the terminal differentiation/maturation of new fat cells. Using the same approach but a different FABP4/aP2-Cre transgene, Jones et al. also found that compared with wild-type animals, mice with adipocyte-specific PPARγ ablation had decreased weight gain, WAT mass, and adipokine levels when fed an HFD, together with WAT inflammation and fibrosis and fatty liver [132]. However, in contrast to He et al., they reported that adipocyte-specific PPARγ knockout mice were completely protected from HFD-induced insulin resistance and impaired glucose tolerance. In euglycemic–hyperinsulinemic clamp studies, the whole-body glucose disposal rate was in fact higher in adipocyte-specific PPARγ knockout versus wild-type mice fed an HFD. This was due to higher glucose disposal at the liver but not the muscle level, as hepatic glycogen synthesis and content were enhanced in knockout versus wild-type mice, whereas muscle glucose uptake was similar in the two groups. These findings suggest a central role for the liver in protecting from whole-body insulin resistance by increasing glucose metabolism despite enhanced fat content, possibly due to a compensatory increase in hepatic PPARγ expression. However, a comparison of different Cre lines suggested that both the mild WAT phenotype and the contrasting data regarding the associated metabolic derangement might be due to the limited specificity and efficacy of the FABP4/aP2-Cre approach in mediating adipocyte-specific recombination [133]. An adipocyte-specific PPARγ knockout mouse model generated using a transgenic Cre line driven by the regulatory region of adiponectin (Adipoq–Cre) was in fact shown to have nearly absent WAT (and BAT) and a small number of adipocytes dispersed among a great number of stromal cells [134]. This was associated with a severe reduction in adipokine levels, massive liver steatosis, dramatic elevation of glucose and insulin levels, and extreme insulin resistance. The finding that selective ablation of PPARγ in mature adipocytes by using a tamoxifen-inducible recombination system caused a rapid and massive death of white and brown *pparg*^−/−^ adipocytes, which were subsequently replaced by newly formed *pparg*^+/+^ adipocytes, confirmed the critical role of PPARγ for the survival of mature adipocytes [135].

#### 2.3.4. Other Factors

Mice knockouts for E2F1 were also found to be resistant to HFD-induced obesity, as the lower increase in body weight versus wild-type mice was entirely explained by a lower increase in WAT mass [95].

In contrast with the in vitro studies showing a compelling role for SREBP-1c in promoting adipogenesis, in vivo studies provided different results. In fact, the surviving SREBP-1 knockout mice (only 15%) showed no change in WAT development or the expression of key adipogenic genes when compared to wild-type mice, suggesting a compensation from the upregulation of SREBP-2 or other factors [136]. Even more surprisingly, transgenic mice overexpressing the nuclear form of SREBP-1c (nSREBP) in adipocytes under the control of the FABP4/Ap2 enhancer/promoter showed a phenotype resembling human congenital generalized lipodystrophy. The differentiation of AT was markedly impaired, with the decreased WAT consisting almost entirely of small immature adipocytes and the increased BAT containing hypertrophic, fat-laden brown adipocytes resembling immature white cells as well as reduced expression of adipogenic genes. This was associated with fatty liver, severe insulin resistance, and impaired glucose homeostasis. A possible explanation for these findings was the interference with adipocyte differentiation by the very high nuclear levels of nSREBP-1c, as the FABP4/aP2-nSREBP-1c transgene encodes a truncated version of SREBP-1c that enters the nucleus directly with no requirement for proteolysis and no feedback regulation [137].

Mice overexpressing PEPCK in adipocytes from the FABP4/aP2 promoter showed increased body weight, fat mass, and adipocyte size as a consequence of increased glycerol 3-phosphate synthesis and FA re-esterification. This was associated with decreased FFA and normal leptin levels in the circulation, which were likely responsible for the preserved insulin sensitivity and glucose tolerance [138].

Mice lacking Schnurri-2 showed a reduced body weight and epidydimal WAT mass with a decrease in the size but not in the number of adipocytes; however, insulin sensitivity and glucose tolerance were increased despite lipodystrophy [139]. These findings are similar to those observed in mice heterozygous for CREB-binding protein, a co-activator for several transcriptions, including SREBPs, C/EBPs, and PPARs, which were also completely protected from HFD-induced obesity [140].

Mice lacking Pref-1/Dlk1 showed accelerated adiposity associated with growth retardation; this phenotype was also observed in mice heterozygous for paternally but not maternally expressed *pref1*-null allele [141]. Mice expressing Pref-1 in adipocytes from the FABP4/aP2 promoter showed severely reduced fat mass due to decreased adipocyte size and reduced expression of adipocyte markers and adipokines, which are associated with increased triglycerides and impaired insulin sensitivity and glucose tolerance [142]. These mice were also found to be resistant to HFD-induced obesity, as they gained less weight and fat mass but showed higher FFA levels and more severe insulin resistance and glucose intolerance than their wild-type counterparts [143].

Mice deficient for KLFs showed no or only a minor phenotype. In fact, heterozygous KLF5 knockout mice exhibited impaired development of WAT characterized by a smaller adipocyte size, which, however, reversed by 4 weeks of age [144], whereas mice lacking KLF13 showed normal adipogenesis [145].

**Table 1 cells-13-00763-t001:** Studies in animal models investigating the role of molecular regulators of adipogenesis.

Study [Reference]	Mouse Model		Normal Diet/Genetic Background		Dietary/Genetic Obesity	
	Molecule	Manipulation	Fat Mass/Cells	Metabolism	↑ Body Weight/Fat Mass	Metabolic Derangement
** *Conserved developmental signaling pathways* **						
Qian et al. [108]	BMP-4	Adipocyte-specific (FABP4) overexpression	↓ WAT mass ↓ size ↑ number	↑ insulin sensitivity ↑ glucose tolerance	↓	↓
		Adipocyte-specific (FABP4) deletion	↑ WAT mass ↑ size ↓ number	↓ insulin sensitivity ↓ glucose tolerance	NA	NA
Soofi et al. [109]	Kielin/chordin-like protein	Global overexpression	↓ WAT mass	= insulin sensitivity = glucose tolerance	↓	↓
		Global deletion	= WAT mass	= insulin sensitivity = glucose tolerance	↑/=	↑
Brown et al. [110]	Follistatin-like 3	Global deletion	↓ fat mass	↑ insulin sensitivity ↑ glucose tolerance	NA	NA
	Follistatin and follistatin-like	Global deletion	↑ fat mass	↓ insulin sensitivity ↓ glucose tolerance	NA	NA
Longo et al. [111]	Wnt10b	Adipocyte-specific (FABP4) overexpression	↓ WAT mass = size ↓ number	↑ insulin sensitivity ↑ glucose tolerance	↓	↓
Wright et al. [112]	Wnt10b	Adipocyte-specific (FABP4) overexpression	↓ WAT mass	NA	↓	↓
Chen et al. [113]	β-catenin	Adipocyte-specific (Adipoq) deletion	↓ WAT mass	NA	↓	↓
Bagchi et al. [114]	Wntless	Adipocyte-specific (Adipoq) deletion	=	=	↓	↓
Mori et al. [115]	SFRP-5	Global deletion	↑ WAT mass ↑ size = number	NA	↓	↓
Geoghegan et al. [116]	TCF7L2	Adipocyte-specific (Adipoq) deletion	NA	NA	↑	↑
Li et al. [117]	Ptc1	Global deletion at the carboxyl-terminal cytoplasmic region	↓ fat mass ↓ size ↓ number	= glucose tolerance	NA	NA
Martin et al. [118]	Hh	Sonic Hh-IgG fusion protein injection	↑ fat mass	NA	NA	NA
Buhman et al. [119]	Hh	Anti-Hh antibody injection	NA	NA	↓	NA
** *CCATT enhancer binding proteins* **						
Tanaka et al. [120]	C/EBPβ	Global deletion	Non-significant ↓ WAT mass	NA	NA	NA
	C/EBPδ	Global deletion	Non-significant ↓ WAT mass	NA	NA	NA
	C/EBPβ and C/EBPδ	Global deletion	Severe ↓ WAT mass	NA	NA	NA
Wang et al. [121]	C/EBPα	Global deletion	Absent WAT	hypoglycemia	NA	NA
Linhart et al. [122]	C/EBPα	Global deletion (except liver)	Absent WAT	↑ insulin = glucose	NA	NA
** *Peroxisome proliferator-activated receptor γ* **						
Duan et al. [124]	PPARγ	Global deletion (except trophoblasts)	Severe ↓ WAT mass	↓ insulin sensitivity ↓ (M) ↑ (F) glucose tolerance	NA	NA
Koutnikova et al. [125]	PPARγ2	Hypomorphic mutation	Severe ↓ WAT mass	mild ↓ insulin sensitivity mild ↓ glucose tolerance	NA	NA
Miles et al. [126]	PPARγ	Global deletion (heterozygous)	= WAT mass	↑ insulin sensitivity	NA	NA
Rosen et al. [85]	PPARγ	Chimera	= WAT mass with only *pparg*^+/+^ cells	NA	NA	NA
Zhang et al. [127]	PPARγ2	Global deletion	↓ WAT mass ↓ size ↓ number	severe ↓ insulin sensitivity (M) = glucose tolerance	↓	NA
Medina-Gomez et al. [128]	PPARγ2	Global deletion	= WAT mass = size = number	↓ insulin sensitivity ↓ glucose tolerance	↑	=
Freedman et al. [129]	PPARγ	Dominant negative mutation L466A	↓ WAT mass ↓ size (small–medium)	=I nsulin sensitivity	↓	↑
Tsai et al. [130]	PPARγ	Dominant negative mutation P465L	↑ WAT mass ↑ size (large)	= insulin sensitivity ↑ glucose tolerance	↑	=
He et al. [131]	PPARγ	Adipocyte-specific (FABP4) overexpression	↓ WAT mass size (very large, small) ↓ number	= insulin sensitivity = glucose tolerance	↓	↑
Jones et al. [132]	PPARγ	Adipocyte-specific (FABP4) overexpression	↓ WAT mass	= insulin sensitivity = glucose tolerance	↓	↓
Wang et al. [134]	PPARγ	Adipocyte-specific (Adipoq) deletion	severe ↓ WAT mass severe ↓ number	severe ↓ insulin sensitivity severe ↓ glucose tolerance	NA	NA
** *Other factors* **						
Fajas et al. [95]	E2F1	Global deletion	NA	NA	↓	NA
Shimano et al. [136]	SREBP-1	Global deletion	= WAT mass	NA	NA	NA
Shimomura et al. [137]	SREBP-1	Global overexpression (nuclear form)	↓ WAT mass ↓ size	↓ insulin sensitivity ↓ glucose tolerance	NA	NA
Franckhauser et al. [138]	PEPCK	Adipocyte-specific (FABP4) overexpression	↑ WAT mass ↑ size	= insulin sensitivity = glucose tolerance	NA	NA
Jin et al. [139]	Schnurri-2	Global deletion	↓ WAT mass ↓ size = number	↑ insulin sensitivity ↑ glucose tolerance	NA	NA
Yamauchi et al. [140]	CREBp	Global deletion (heterozygous)	↓ WAT mass	↑ glucose tolerance	↓	↓
Moon et al. [141]	Pref-1/Dlk1	Global deletion	↑ WAT mass ↑ size	NA	NA	NA
Lee et al. [142]	Pref-1/Dlk1	Adipocyte-specific (FABP4) overexpression	↓ WAT mass ↓ size	↓ insulin sensitivity ↓ glucose tolerance	NA	NA
Villena et al. [143]	Pref-1/Dlk1	Adipocyte-specific (FABP4) overexpression	NA	NA	↓	↑
Oishi et al. [144]	KLF5	Global deletion	↓ WAT mass ↓ size (transient)	NA	NA	NA
Gordon et al. [145]	KLF13	Global deletion	= fat mass	NA	NA	NA
Yang et al. [106]	Galectin 12	Global deletion	↓ WAT mass ↓ size = number	NA	↓	↓
Baek et al. [100]	Galectin 1	Global deletion	= WAT mass = size	= glucose	↓	↓
Baek et al. [101]	Galectin 3	Global deletion	↓ WAT mass ↓ size	= glucose	↓	↓
Blasetti Fantauzzi et al. [102]	Galectin 3	Global deletion	= WAT mass ↓ size ↑ number	↓ insulin sensitivity (transient) ↓ glucose tolerance	NA	NA
Pejnovic et al. [146]	Galectin 3	Global deletion	= WAT mass	= insulin sensitivity = glucose tolerance	↑	↑
Li et al. [147]	Galectin 3	Global deletion	= WAT mass	↑ insulin sensitivity ↑ glucose tolerance	=	↓
Pang et al. [148]	Galectin 3	Global deletion	↑ WAT mass	↑ glucose	↑	↑
Takada et al. [149]	Galectin 3	Global deletion	↓ muscle fat	NA	NA	NA
Martínez-Martínez et al. [150]	Galectin 3	Inhibitor (modified citrus pectin) injection	= WAT mass	NA	=	NA

BMP = bone morphogenetic protein; Wnt = wingless-type; SFRP = secreted frizzled-related protein; TCF7L2 = transcription factor 7-like 2; Ptc = Patched; Hh = Hedgehog; C/EBP = CCATT enhancer binding protein; PPAR = peroxisome proliferator-activated receptor; E2F = E2 transcription factor; SREBP = sterol regulatory element-binding protein; PEPCK = phosphoenolpyruvate carboxykinase; CREBp = phosphorylated cyclic AMP response element-binding protein; Pref = preadipocyte factor; Dlk = delta-like non-canonical notch ligand; KLF = Krupel-like factor.

Mice knockouts for galectin-12, which is preferentially expressed in adipocytes and localized to lipid droplets, were shown to have a similar body weight but substantially lower sWAT and vWAT mass and triglyceride content compared with wild-type mice due to a lower size but not number of adipocytes. Moreover, *Lgals12*^−/−^ mice exhibited significant protection from dietary and genetic obesity, as shown by the lower weight gain, insulin resistance, and glucose intolerance compared to *Lgals12*^+/+^ animals, consistent with increased lipolysis enhancing mitochondrial respiration and whole-body energy expenditure [106]. No effect on body weight, adiposity, or metabolism was reported with global deletion of galectin-1, which is expressed in several tissues, including adipocytes, where it localizes to the nucleus during differentiation. However, when fed an HFD, *Lgals1*^−/−^ mice showed significantly lower increases in body weight, WAT mass, adipocyte size, and expression of lipogenic and inflammatory genes and glucose but not FFA or triglyceride levels compared with *Lgals1*^+/+^ animals, consistent with protection from dietary obesity [100]. At variance with galectin-12 and -1, several studies have investigated the effect of ablation or inhibition of galectin-3, which shows a widespread expression in the body and serves multiple functions, thus possibly explaining the somewhat contrasting data reported on WAT morphology and function in mice fed standard chow or an HFD. Both body weight and sWAT and/or vWAT mass were consistently found to be normal in young *Lgals3*^−/−^ mice under standard chow conditions [102,146,147], except in one study showing higher fat mass at 20 weeks of age [148]. Blasetti Fantauzzi et al. showed that the unchanged WAT mass at 2 months of age was the result of an increased cell number compensating for a reduced size of adipocytes, which showed signs of immaturity, such as decreased expression of genes coding for adipogenic transcription factors, lipogenic enzymes, and adipokines, indicating impaired terminal differentiation [102]. Moreover, despite no sign of adipocyte degeneration/death or fat diversion to ectopic sites, this WAT phenotype was associated with tissue and systemic inflammation and insulin resistance but normal glucose levels, which, however, became elevated one month later [102]. These findings suggested that adipocyte immaturity may directly cause metabolic derangement, i.e., in the absence of an increased demand for fat storage and, hence, independently of increased fat accumulation and redistribution. This interpretation was supported by the finding that maturation of adipocytes was observed in *Lgals3*^−/−^ mice at 5 months of age, along with resolution of inflammation and restoration of insulin sensitivity, though glucose levels remained abnormal due to reduced insulin secretion, suggesting that ablation of galectin-3 also enhances age-dependent impairment in β-cell functional reserve [102]. Pang et al. also found higher blood glucose and glycated hemoglobin levels in *Lgals3*^−/−^ versus *Lgals3*^+/+^ mice on a standard diet, suggesting that galectin-3 modulates β-cell function [148]. In contrast, decreased body weight, WAT mass, and adipogenic and lipogenic gene expression were observed in 17-month-old *Lgals3*^−/−^ mice on a standard chow, without any sign of inflammation or abnormal glucose homeostasis [101]. Moreover, Li et al. found that 8-month-old *Lgals3*^−/−^ mice had better insulin sensitivity and glucose tolerance than *Lgals3*^+/+^ animals despite similar body weight, suggesting that (macrophage-derived) galectin-3 impairs insulin action, as also shown by the in vitro finding of direct galactin-3 binding to the insulin receptor and inhibition of downstream signaling [147]. Finally, mice knockouts for galectin-3 showed suppression of intermuscular AT formation during muscle regeneration, consistent with the observation that this lectin promotes adipogenic differentiation of MSCs in muscle, which is a mechanism by which obesity favors the development of sarcopenia [149]. However, response to HFD was also very different among studies, despite the fact that in three out of four of those testing this response the source and background of the *Lgals3*^−/−^ mice were the same [101,146,147,148]. In particular, mice knockouts for galectin-3 or treated with an inhibitor of this lectin were found to have increased [146,148], unchanged [147,150], or decreased [101] weight gain and WAT mas, associated with increased [146,148] or decreased [150] inflammation and impaired [146,148], unchanged [101], or improved [147] insulin resistance and/or glucose tolerance. However, these contrasting results might be, at least in part, attributed to the use of models of global inhibition of galectin-3, either genetic or pharmacological, and hence the lack of other functions of this lectin, involving the regulation of β-cell function and the immune system [151] and the advanced glycation end-product (AGE)-receptor function exerted by this lectin, which is opposite to that of the receptor for AGEs (RAGE) [152].

#### 2.3.5. Considerations from Animals Studies

The results of the above studies in animals lacking or overexpressing factors known to exert a stimulatory or inhibitory effect on adipogenesis showed that defects in this process translated into a wide range of phenotypes, which could be detected under standard conditions or in response to overfeeding using models of dietary or genetic obesity.

Apart from the opposite results obtained when increasing or decreasing the same factor using different approaches or compared with the in vitro studies, which may be attributed to methodological reasons, the sequence of events linking defective adipogenesis to metabolic derangement through the predominance of the hypertrophic over the hyperplastic WAT response was not consistently observed. In fact, mice showing impaired WAT development had either unchanged or reduced body weight along with a decrease in WAT mass of various extent, which was due to a reduction in adipocyte numbers and/or size. This phenotype resembling human lipodystrophy was protected from dietary or genetic obesity, as body weight gain was less, but the reduced WAT capacity to accommodate excess fat was associated with not only impaired, as expected, but unchanged or even improved insulin sensitivity and glucose tolerance. In some of these studies, the favourable metabolic profile was attributed to various factors, including increased energy expenditure, decreased inflammatory cytokines, increased leptin or decreased resistin levels, or compensation by the liver or muscle. Conversely, in some other instances, a normal WAT mass in the absence of fat diversion to ectopic sites was associated with significant metabolic derangement.

Taken together, these findings highlight the extreme complexity and heterogeneity of defects of adipogenesis, which may end up with different phenotypes in terms of WAT mass and adipocyte number and size, depending on the type of defect and the stage at which the process is blocked or retarded. In addition, the discrepancies between the WAT phenotype and the metabolic profile point to a role for additional factors operating in excess fat disposal.

## 3. Impaired White Adipose Tissue Remodeling in Aging

Serving as a crucial energy reservoir and hormonal regulator vital for systemic energy metabolism and immune homeostasis, WAT plays an active role in the aging process and the development of age-related illnesses [3,153]. Consistently, interventions aimed at delaying or preventing adipocyte dysfunction have shown potential for extending both health span and lifespan [154,155].

### 3.1. Significance of Changes in White Adipose Tissue Distribution, Cellular and Lipid Composition, and Molecular Signature in Aging

Appropriate remodeling of WAT with aging is important for maintaining health and function. Research conducted in humans, especially among healthy centenarians, has revealed the critical role of WAT in extreme old age [156]. This tissue secretes various circulating factors, including adiponectin, known for their association with a protective metabolic and anti-inflammatory profile during aging [157,158]. Furthermore, research conducted on mice infected with bacteria has demonstrated that WAT serves as a reservoir for memory T cell populations, facilitating a protective memory response to infections and thereby contributing to the immune system’s defense mechanisms [159]. On the other hand, there is experimental and clinical evidence suggesting that changes in WAT with advancing age may contribute to age-related dysfunctions in other organs and systems, rather than merely serving as a marker of WAT aging (Figure 3).

First of all, aging is associated with the redistribution of WAT. Subcutaneous WAT experiences a general decline as individuals age, particularly in the abdomen and limb region [160,161]. The decrease in both the number and differentiation potential of adipocyte precursors cells (APCs) contributes to this phenomenon [162]. In parallel, there is an accumulation of visceral fat and intramuscular fat. These changes are linked to the development of the metabolic syndrome accompanied by an increased risk of T2D and CVD during aging [163]. Notably, middle-aged mice already display impaired plasticity in sWAT, which could contribute to the early onset of insulin resistance [164]. Furthermore, the accumulation of T cells and B cells, which has been demonstrated as a typical feature of aging in WAT, becomes noticeable from middle age [165]. The process of aging is further linked to changes in the balance of macrophages within AT, favoring a shift towards a pro-inflammatory state. This is indicated by the decline in the relative abundance of M2-like macrophages compared to M1-like macrophages, which is connected to a reduction in the expression of PPARγ [166]. Overall, besides anticipating aging-related changes in other tissues [165], the inflammatory cell signature observed during the physiological aging of WAT is unique and differs from that induced in the context of diet-induced obesity [166].

From a molecular standpoint, targeted proteomic analysis found significant age-related changes in murine WAT, including shifts in lipid and central carbon metabolism, electron transport chain complexes, and inflammation [167]. RNA sequencing of 17 organs confirmed that WAT ages early, with increased expression of certain age-related genes during midlife, preceding similar changes in other organs. This heightened gene expression is associated with immune responses and cytokine-mediated inflammation [165]. WAT was also identified as a significant source of age-associated plasma proteins, potentially accelerating aging throughout the body [165]. It follows that interventions targeting WAT aging may help suppress age-related diseases and prolong the lifespan.

Finally, lipidomic research exploring the alteration of WAT in mice as they age reveals substantial modifications in lipid composition as well. These changes encompass modifications in both phospholipid and sphingolipid profiles within WAT [168,169]. Specifically, as observed in the gonadal WAT of 24-month-old mice, aging leads to elevated levels of phosphatidylethanolamine, phosphatidylcholine, phosphatidylinositol, and ether-phospholipids [169]. Regarding sphingolipids, the increased activity of enzymes involved in ceramide production, such as serine palmitoyl transferase and ceramide synthase, likely contributes to the rise in ceramide species like ceramide 16:0 and 20:0 observed in the WAT of aging mice [170]. The overall levels of sphingomyelin also undergo alterations during aging, with an increase in abundance observed within WAT of rats aged 24 months [170]. Changes in the lipid composition of aging WAT in rodent models have been linked to heightened vulnerability to oxidative stress [171], reduced fluidity of cell membranes [172], and compromised thermogenic capability due to impaired differentiation into beige/brown adipocytes [168]. Significantly, changes in the profiles of phospholipids and sphingolipids detected in WAT of rodents seem to mirror changes observed in human plasma [173,174]. This suggests that the remodeling of lipid composition in WAT during aging substantially contributes to the observed variations in plasma lipid composition [175]. In human studies, elevated plasma concentrations of sphingolipids have been suggested as potential indicators for insulin resistance and Alzheimer’s disease (AD) [176,177]. Moreover, ceramide 16:0 has been singled out as a promising biomarker for aging [178]. Together, these findings suggest that plasma lipid composition may serve as a biomarker of age-related changes in AT and an indicator of healthy or pathological aging.

### 3.2. Cellular Senescence, Autophagy, and Fibrosis of White Adipose Tissue in Aging

Cellular senescence is characterized by cells entering a state of permanent cell cycle arrest while remaining metabolically active, alive, and resistant to apoptosis [179]. While cellular senescence serves beneficial functions in tissue development, repair processes, and tumor suppression, senescence in adipocytes leads to various dysfunctions. These include compromised lipid handling, abnormal production of adipocytokines, insulin resistance, and hindered adaptive thermogenesis [180].

Senescent adipocytes, aside from their inability to divide, develop what is known as a senescence-associated secretory phenotype (SASP), characterized by the persistent secretion of elevated levels of extracellular matrix-degrading enzymes, pro-inflammatory cytokines, and immune modulators, such as tumor necrosis factor-alpha (TNFα), interleukin (IL)-6, and IL-8 [181]. Thus, SASP plays a role in the development of localized and systemic chronic low-grade inflammation, known as inflammaging, through autocrine and paracrine mechanisms [182,183,184]. Additionally, the capacity of senescent APCs to differentiate into functional mature white, beige, or brown adipocytes decreases with aging [183,185,186,187,188,189]. In particular, the acquisition of a SASP phenotype during the aging process contributes to the inability to induce WAT beiging in response to cold temperatures. This hindrance can be reversed by genetic or pharmacological means aimed at targeting the p38/MAPK-p16^Ink4a^ pathway in aged mouse and human beige progenitor cells [185].

Both sWAT and vWAT WAT exhibit varying correlations with aging-related changes, including cellular senescence. Telomere length, a marker indicating the risk of cellular senescence, consistently appears shorter in sWAT compared to vWAT, implying that sWAT is more susceptible to age-related damage [190]. Recently, a subset of aging-dependent regulatory cells, termed ARCs, was identified in peripheral sWAT of aged mice [191]. These cells, characterized by markers typical of APCs, such as platelet-derived growth factor receptor α (PDGFRα) and Pref-1, also exhibited elevated expression of inflammatory markers, including CD163, Adgre1 (F4/80), and Nuclear Factor Kappa B Subunit 1, with heightened levels observed in the oldest cohort of mice [191]. Notably, ARCs also exhibit elevated PU.1 expression, a transcription factor encoded by the Spi1 gene, which is known to inhibit preadipocyte differentiation by suppressing PPAR-γ [192]. Therefore, in addition to their inability to differentiate, ARCs exhibit a secretory profile that hinders the differentiation of nearby adipose progenitors, ultimately contributing to age-related sWAT loss.

Autophagy, a dysregulated process in senescent cells [193], has shown associations with various features of aging WAT. These include but are not limited to blunted adipogenesis, reduced adipocyte functionality, inflammatory reactions, and fibrosis. However, the precise role of autophagy in regulating lipid metabolism, sustaining energy balance, and impacting insulin sensitivity remains incompletely understood. In contrast to the widespread reduction in autophagic activity associated with aging, which leads to disruptions of homeostasis in several tissues, aged adipocytes exhibit increased autophagy. This augmentation is attributed to diminished levels of Rubicon, a protein known to negatively regulate autophagy [194]. Increased autophagic activity in aged adipocytes suppresses PPARγ by promoting the degradation of Steroid Receptor Coactivator-1 and Transcriptional Intermediary Factor 2, essential coactivators of this master regulator of adipocyte differentiation, lipogenesis, function, and insulin sensitivity [195]. Consistently, a reduction in Rubicon leads to adverse outcomes, such as glucose intolerance and the accumulation of fat in the liver [194].

Another feature of aging WAT is fibrosis, characterized by an excessive deposition of extracellular matrix components. This process disrupts tissue architecture and reduces the WAT’s capacity to accommodate lipids. In general, fibrosis is a signature of unhealthy WAT [196]. Within WAT, myofibroblasts with fibrosis-inducing properties develop from APCs expressing PDGFRα, which typically undergo differentiation into beige adipocytes [197,198]. Recent experimental findings indicate that the fibrogenic-to-adipogenic transition of APCs is compromised in aging WAT. While mature adipocytes from WAT of young mice secrete β-hydroxybutyrate to promote the differentiation of APCs into beige adipocytes and suppress fibrosis, the age-related reduction of the transcription factor PRDM16 is responsible for the decreased secretion of this ketone body by adipocytes [192]. Consequently, this leads to the promotion of WAT fibrosis and the suppression of beige fat formation. Notably, targeting fat cells by increasing PRDM16 expression [192] or by promoting the formation of a complex with the cold-inducible transcription factor GTF2IRD1 [199] is sufficient to enhance systemic glucose homeostasis and WAT insulin sensitivity through the repression of WAT fibrosis. Importantly, these improvements occur regardless of changes in body mass [199].

To summarize, all of these findings indicate that the molecular and structural changes that WAT undergoes during aging, including senescence, abnormal autophagy, and the fibrosis process, play a role in systemic dysfunction, such as inflammaging and glucose homeostasis alterations, rather than merely being a consequence of adipose maladaptation.

### 3.3. Potential Therapeutic Interventions for Age-Related Adipocyte Dysfunction

The potential to delay the processes that characterize the aging of WAT with drugs capable of correcting age-associated adipocyte dysfunction is an area of increasing interest. Several studies employing emerging or established systemic therapies have produced intriguing results in mitigating age-related changes in the adipose organ (Figure 4), which have been associated with lifespan extension in certain experimental studies.

The removal of senescent cells by using senolytics has been demonstrated to delay age-related decline and the progression of chronic diseases in both animal models and humans [200,201,202,203]. A recent discovery has identified the glycoprotein nonmetastatic melanoma protein B as a key molecular target for senolytic treatment. Indeed, vaccination aimed at this seno-antigen has been shown to diminish senescence in WAT and ameliorate age-related pathological features in mice [204].

Even the experimental use of conventional medications prescribed for managing T2D has shown interesting effects on the aging process of WAT. Biguanides have long been suggested to offer advantages for both health span and life span in experimental research [205]. Interestingly, metformin has been observed to enhance adipogenesis, glucose uptake, and mitochondrial fatty acid oxidation in adipose stromal cells obtained from the sWAT of women aged over 55 years by activating AMP-activated protein kinase [205]. Additional mechanisms for the beneficial effects of metformin on WAT aging are represented by the improvement of PPARγ and sterol regulatory element-binding protein signaling as well as collagen trimerization in WAT [206].

Finally, thiazolidinediones (TZDs) are known to improve WAT and whole-body insulin sensitivity by acting as ligands for PPARγ, thereby promoting adipocyte differentiation, enhancing adipocyte function, and reducing the release of free fatty acids into circulation [207,208]. A recent study has revealed that administering low doses of TZDs led to improvements in age-related decline of WAT. In particular, TZDs were found to decrease inflammation and fibrosis in aging WAT, thereby playing a role in preserving overall AT homeostasis [209].

### 3.4. White Adipose Tissue Maladaptation in Age-Related Disorders

In addition to the increased cardiometabolic risk, there is evidence suggesting that WAT maladaptation influences typical pathologies affecting the elderly by disrupting metabolic homeostasis and the regenerative capabilities of other organs and tissues [163]. For instance, the accumulation of ectopic fat in various tissues due to the diminished function of WAT, including the muscle, liver, dermis, and bone marrow, is associated with age-related disorders, such as sarcopenia, hepatic steatosis, diminished dermal elasticity, and osteoporosis [187,210,211,212,213]. In the following sections, we will explore WAT maladaptation in common pathologies of the elderly, including osteoarthritis (OA), osteoporosis, cognitive decline and dementia, and cancer cachexia (CC) (Figure 4). We will examine its potential role as both a contributor to and a target of chronic and cancer processes, suggesting it as a therapeutic target to mitigate the progression of frailty.

#### 3.4.1. Osteoarthritis

OA stands as the predominant type of arthritis and is acknowledged as a primary source of pain and disability [214,215]. Its prevalence encompasses 3.3% of the global population, a statistic as of 2015 [216,217], with a tendency to increase with age. Among individuals aged 60 and above, approximately 10% of males and 18% of females experience this condition [218].

The risk of OA is increased among individuals who are overweight. However, emerging evidence suggests that alterations in biomechanical joint loading resulting from higher body mass are insufficient to fully explain the severity of knee OA associated with obesity [214,219,220,221,222]. These findings imply that additional factors associated with dysfunction in AT, primarily WAT, including abnormal release of adipokines, inflammation, and ensuing metabolic disorders, may significantly contribute to this process [214,215,220,223,224]. This topic is of utmost importance, especially considering the current lack of disease-modifying drugs for OA.

To support the hypothesis that WAT contributes to OA, a recent study has demonstrated that transgenic mice exhibiting lipodystrophy, regardless of whether they were fed a standard chow or a high-fat diet, exhibit resistance to both spontaneous and post-traumatic OA. This resistance persists despite the presence of numerous factors formerly associated with OA, including body weight and inflammation. Additionally, the implantation of a mixture of sWAT and vWAT from wild-type mice reintroduced susceptibility to cartilage damage and post-traumatic OA in this preclinical animal model of lipodystrophy [225]. This indicates a direct correlation between WAT and cartilage degradation, irrespective of body weight, and implies that paracrine signaling from WAT contributes to joint degeneration.

Previous and subsequent studies confirm a correlation between AT and joint health, elucidating insights into the natural history of OA and providing clues for treatment. Of interest is the observation that the therapeutic efficacy of injectable WAT-derived products for OA treatment could depend on the health status of the adipose organ and the metabolic condition of the patient [226]. Injectable therapies utilizing WAT-derived products have shown potential in modifying disease outcomes in joint tissues according to preclinical investigations conducted on animal models of OA [227]. However, limited clinical evidence exists for this autologous WAT-derived cell therapy. While some studies suggest safety and benefits in joint function and pain relief [228,229,230], findings from high-quality trials often vary widely, frequently resulting in unsatisfactory clinical outcomes [231,232,233]. The variability in results can be attributed directly to the substantial variances in the quality of WAT utilized in the preparation of injectable products [234]. Dysfunctional WAT from patients with obesity and OA, characterized by senescence, abnormal adipokine production, inflammation, and fibrosis may be the reason for the unsatisfactory clinical outcomes of this cell-based treatment [226]. This suggests that products derived from WAT of patients with obesity may possess inadequate properties for treating OA. Similarly, advancing age has been recognized as a significant risk factor for the ineffectiveness of autologous WAT-derived cell therapy [235], owing to its influence on the paracrine and inflammatory functions of WAT-derived mesenchymal stem cells [236].

Finally, age-related changes in the infrapatellar fat pad may also play a role in knee OA [237,238]. Lower total collagen deposition and elastic fibers around the adipocytes were found in elderly donors compared to young donors [239]. Combined with the finding that the infrapatellar fat pad serves as a significant source of IL-6 and other inflammatory modulators [237] and is affected by fibrotic and vascular changes in OA patients [238], this local WAT depot has attracted attention for its active involvement in joint degeneration. However, further investigation is required to explore the functions of the various WAT depots in OA, particularly concerning the impact of adipose-derived stem cells from both healthy and unhealthy WAT on OA.

#### 3.4.2. Senile Osteoporosis

Osteoporosis is a metabolic bone condition characterized by a gradual decline in bone mass, microstructure, and biomechanical properties. This results in a weakening of bone strength, increasing the likelihood of fractures, particularly from minor impacts (i.e., brittle fractures) [240]. The relation between body fat and the skeleton is as complex as it is contentious. Each type of AT, including BAT, sWAT, and particularly vWAT, has been associated with variations in bone mineral density (BMD) and microarchitecture [241]. On the one hand, obesity-related excess vWAT alters adipokine secretion, increasing pro-inflammatory cytokines that may disrupt bone remodeling [242]. Consistently, an inverse correlation was observed between spinal cortical and trabecular BMD and vWAT content, even after adjusting for BMI [243]. On the other hand, increased mechanical loading resulting from elevated BMI has been linked to higher BMD and improved microstructure, thus positively impacting both cortical and trabecular bone health [244]. To reconcile conflicting results, a U-shaped correlation between visceral WAT and BMD has been proposed [245], thus further increasing the level of complexity.

In addition to WAT and BAT, bone marrow AT (BMAT) has gained recognition as an extra fat reservoir involved in bone-fat interaction. It has been suggested to be associated with BMD loss during aging (i.e., senile osteoporosis) [246,247]. From a morphological and functional perspective, the fat cells constituting the BMAT, with their unilocular appearance and the production of adipokines, such as leptin and adiponectin [248], exhibit characteristics more akin to white adipocytes [249,250]. BMAT could constitute approximately 5% of the total fat mass in adult individuals. Bone marrow adiposity increases with age, occupying about 50% of the bone marrow volume at the age of 25 [251]. It further increases throughout adulthood because of the conversion of bone marrow to BMAT [247,252]. Importantly, while BMAT formation is distinct from that of other fat depots, it is conceivable that BMAT accretion may not be entirely independent of the distribution of WAT, particularly when sWAT redistribution occurs during aging [160,161,253]. Interestingly, the buildup of BMAT is linked to a reduction in bone formation in its vicinity, leading to a decline in bone mechanical strength [254,255].

In senile osteoporosis, bone loss primary results from an altered bone remodeling with either decreased number and/or mineralizing function of osteoblasts, thereby resulting in a reduction in bone mechanical strength and eventually an increased risk of fractures [256]. There is experimental evidence suggesting that WAT, including BMAT, could contribute to bone loss through mechanisms involving lipotoxicity, impairing osteoblast function and survival [257,258,259]. Several animal models of progeria exhibit osteoporosis [260], including the senescence accelerated prone 8 (SAMP8) mouse strain, which mimics clinical features of aging in humans, including senile osteoporosis [261,262]. Interestingly, lifelong dietary supplementation with fish oil, rich in ω3 (n-3) fatty acids, has been shown to partially prevent age-related bone loss in mice [262]. This effect was associated with reduced volume of BMAT [263]. The study did not investigate the mechanisms by which fish oil aids in osteoblast function and bone health, while also inhibiting BMAT expansion. The authors suggest that it may suppress lipotoxicity resulting from age-related BMAT accumulation, in which fatty acid composition, high in saturated fatty acids, is emerging as a key signature of this fat depot compared to other WAT depots [247]. This hypothesis finds partial support in prior research demonstrating fish oil’s ability to reduce PPAR-γ expression, thereby promoting stem cell differentiation towards osteoblastic rather than adipogenic lineages [262].

To conclude, besides the necessity of conducting well-designed clinical trials to acquire robust and conclusive data on the relationship between WAT and bone, further research must be undertaken to validate whether BMAT can truly be regarded as a robust and reliable indicator of bone health and fracture risk when compared to other fat depots. This confirmation could potentially reveal new clinical perspectives in the management of age-related osteoporosis and, possibly, bone loss associated with metabolic conditions [264].

#### 3.4.3. Cognitive Decline and Dementia

As our societies age, the prevalence of dementia worldwide is projected to increase significantly, with estimates suggesting a rise from 55 million individuals affected in 2019 to a staggering 139 million by 2050 [265]. Concurrently, the associated expenses are anticipated to surpass a twofold increase over the same timeframe.

An important interplay exists between AT and the brain. It is well known that the brain influences various aspects of AT function, including lipolysis [266], cytokine production [267], and, through neuroimmune interactions, plays a role in thermoregulation [268]. However, several hints suggest that the reverse may also be true. In particular, a great deal of research indicates that AT influences brain function via adipokines, which can cross the blood-brain barrier to affect target areas or regulate its function. Adipokines play a significant role in various brain functions, including synaptic plasticity, memory formation, neurogenesis, inflammatory responses, and the maintenance of the blood-brain barrier [269]. Consistently, obesity, T2DM, and other metabolic disorders associated with altered adipokine production increase the risk of cognitive impairment and numerous neurodegenerative diseases, including AD [270,271,272,273]. Accordingly, adiposity has been proposed as an independent factor favoring the development of AD [274].

However, as observed with osteoporosis, the connection between obesity and dementia, particularly AD, is intricate and controversial [275]. On one hand, magnetic resonance imaging and computed tomography research revealed brain structure changes in overweight/obese individuals [276,277]. Further studies found a link between visceral WAT and decreased brain volume in middle-aged adults, particularly impacting the temporal lobe and hippocampus size [278]. Additionally, a connection has been established linking obesity, neuroinflammation, autoimmune diseases, such as multiple sclerosis, and neurodegenerative disorders, including AD and Parkinson’s disease [279,280,281,282,283,284,285]. Contrary to these studies, other research suggests that individuals classified as underweight (BMI < 20 kg/m^2^) show a heightened susceptibility to dementia, while those categorized as severely obese (BMI > 40 kg/m^2^) have a decreased risk of dementia compared to those with a healthy BMI [275,286]. Interestingly, a more significant decrease in BMI during aging, as well as a low BMI in later life, have been associated with an increased risk of dementia and AD [286]. These results underscore the significance of considering a life-course perspective when examining the relationship between BMI and cognitive function. They also suggest that the inappropriate expression of factors derived from AT, either due to dysfunction in adipocytes caused by body weight issues or the unhealthy remodeling of WAT during aging, could potentially disrupt brain homeostasis and functions.

The role of adipokines in regulating adipose-brain cross-talk and the associated risk of developing cognitive decline and dementia has recently been comprehensively reviewed [269]. However, adipokines are not the sole adipose factor capable of regulating the brain. Preclinical studies have established an association between visceral WAT and impaired memory in obese mice through IL-1 microglial activation, mediated by the NLR family Pyrin Domain Containing 3 [287]. Furthermore, genetically modified mice engineered to release a 20-amino acid peptide called NaKtide in adipocytes demonstrated enhanced hippocampal memory by inhibiting Na,K-ATPase signaling [288]. Notably, recent research in humans has uncovered previously unidentified connections between the transcriptome of WAT and brain function. In particular, RNA sequencing of WAT identified genes linked to cognitive domains in human cohorts. Among them, solute carrier family 18 member A2 (SLC18A2) and regulating synaptic membrane exocytosis 1 (RIMS1) mRNA in WAT were examined for their functional relevance. Briefly, conditional knockdown of SLC18A2 in AT reversed memory impairment induced by HFD in mice. Furthermore, downregulation of Vmat (the Drosophila orthologue of SLC18A2) improved short-term memory, while overexpression of Rim (RIMS1 orthologue) enhanced learning abilities in Drosophila [289]. In the search for accessible biomarkers with prognostic and predictive value, gene expression in peripheral blood mononuclear cells was assessed, revealing that genes associated with cognition in AT (NUDT2, AMPH, UNC5B, OAT, EZR, and NR4A2) exhibited similar associations with cognitive traits across 816 subjects [289]. These findings reinforce the concept of targeting AT as a therapeutic strategy for cognitive dysfunction, by identifying potential biomarkers and clinically relevant therapeutic targets. Delivering miRNAs targeted at AT and influencing cognitive functions appears to offer a safer approach compared to directly targeting the brain.

Unfortunately, this study [289] solely concentrated on individuals with severe obesity (BMI > 35 kg/m^2^), thereby limiting its generalizability. To bolster these findings and broaden their applicability regarding the association between fat and cognitive dysfunction during aging, further research utilizing larger, longitudinal studies is imperative, especially with regards to older adults.

#### 3.4.4. Cancer Cachexia

In addition to excess fat, conditions characterized by the deficiency of WAT, such as cachexia, are associated with notable metabolic issues including insulin resistance, impaired glucose tolerance, and inflammation [290]. This suggests similar or even identical molecular mechanisms linking obesity and scarcity of WAT with metabolic conditions [291], likely reflecting comparable dysfunctions of WAT in conditions of both extreme abundance and deficiency.

Cachexia is a progressive multifactorial condition marked by the loss of skeletal muscle and adipose depots, which does not respond to nutritional interventions [292]. Wasting can result from a variety of chronic conditions such as kidney disease, heart failure, and HIV infection [293]. Significantly, cachexia affects 30% of patients with cancer, contributing to approximately 20% of cancer-related deaths [294]. Originally believed to be solely a result of the increased energy demands of the tumor, the understanding of CC has evolved to recognize it as the outcome of tumor-secreted factors inducing a cachectic metabolism favoring catabolic processes [295]. Another paradigm that has evolved over time is that, while muscle wasting remains a significant aspect of the neoplastic cachectic state, the depletion of WAT may play a decisive role, particularly after the observation that the reduction in fat mass precedes that of lean mass in the progression of CC [292,296], and the demonstration of a direct link between fat breakdown and muscle wasting [297].

By demonstrating elevated gene expression of uncoupling protein 1 and browning markers in cachectic WAT of murine models of lung, liver, and pancreatic cancers, it has been proposed that browning-induced increased energy expenditure contributes to muscle wasting in CC [290]. In particular, Kir et al. [298] have shown that blocking parathyroidhormone-related protein (PTHrP) on adipose tissue prevented CC in mice treated with the cachexia-inducing Lewis lung cancer cells. Nevertheless, heightened resting energy expenditure is not frequently noted in CC, in either patients or mouse models [299,300]. This may suggest a distinct function of PTHrP in CC, separate from its involvement in stimulating the thermogenic process of adipose tissue.

From another perspective, peculiar changes in fat cell metabolism were described in CC. Generally, the reduction in fat mass induced by nutrient deficiency (e.g., fasting) is associated with an increase in lipolysis and a reduction in lipogenesis [301]. However, CC appears to deviate from this general law of balance between lipogenesis and lipolysis when determining fat loss. In fact, while increased WAT lipolysis has definitely been recognized as a key element in the development of CC, some recent evidence suggests that, contrary to previous findings [302,303], lipogenesis is also increased in CC [300,304]. Regarding lipolysis, the main enzymes responsible for the hydrolysis of triacylglycerols and diacylglycerols, ATGL and HSL, respectively, are increased in patients with CC [297]. In alignment with the pivotal role of lipolysis in CC progression, the ablation of HSL and, particularly, ATGL has been demonstrated to protect mice against the loss of WAT and, to some extent, muscle, even in the presence of unchanged levels of other cachexokines, such as TNF-α and IL-6 [297]. Accordingly, therapies targeting adipocyte lipolysis, particularly the ATGL enzyme, have been proposed for the treatment of CC [305]. However, increased lipolysis is not the only metabolic change affecting the adipocyte in CC, and WAT wasting could actually be a result of more complex metabolic rearrangements. In particular, increased lipogenesis has also been reported [300,304]. Increased triglyceride synthesis, along with a high rate of lipolysis, indicates elevated substrate cycling, implying a rapid turnover of the fat mass and ongoing remodeling of the WAT depots. This process necessitates heightened energy demand, potentially resulting in a net loss of WAT. Consistently, exposure to tumor cells and tumor growth in mice has been shown to trigger a futile energy-depleting cycle in cultured white adipocytes and WAT, respectively, resulting in a reduction of WAT ATP levels by more than 50% [300].

In addition to the findings from preclinical models [297], in vitro studies also indicate that a direct effect of tumor-derived factors, rather than indirect effects, is responsible for cachectic fat loss [305]. Recently, compelling evidence supporting the cancer’s ability to induce WAT wasting by impacting the angiocrine signals of WAT endothelium has been generated [305]. As demonstrated using human and mouse cancer models, tumors alter the transcription of endothelial genes in WAT even in the precachectic state, leading to an overactivation of Notch1 signaling in distant WAT endothelium. This subsequently exacerbates tumor-induced WAT loss. Mechanistically, Notch1 signaling increases the synthesis of retinoic acid and IL-33 in endothelial cells, and pharmacological inhibition of retinoic acid signaling was sufficient to prevent WAT wasting in a mouse CC model [306].

To summarize, the relationship between cancer and WAT in CC has not been fully defined. Recent data indicate the need to direct research efforts towards identifying tumor factors that may regulate WAT remodeling, in order to pinpoint targetable pathways to inhibit the progression of tissue wasting.

## 4. Conclusions

The adipose organ has a unique ability to adapt and respond to stimuli, such as energy surplus, by remodeling and expanding through hypertrophy and hyperplasia. While enlargement of existing adipocytes (hypertrophy) beyond a certain threshold is detrimental, de novo adipogenesis, i.e., recruitment of new adipocytes from uncommitted precursors (hyperplasia), allows for accommodating large amounts of lipids without compromising metabolic health. Therefore, adipocyte size is considered a marker of subcutaneous WAT hypertrophy and dysfunction and a predictor of increased cardiometabolic risk. However, studies in individuals with obesity and T2D have shown a bimodal cell size distribution in WAT, with about the same proportion of large and small adipocytes, suggesting that a defect may occur at any stage of the process in which MSCs commit to the adipose lineage to become preadipocytes, which then differentiate into adipocytes. The extreme complexity and heterogeneity of defects of adipogenesis may explain the different phenotypes observed in animal models and humans in terms of WAT mass and adipocyte numbers and size. Additional factors play a role in modulating excess fat disposal resulting in the metabolic derangement associated with WAT dysfunction.

Both an excess and scarcity of WAT can pose health threats to the elderly. The remodeling of WAT during aging is a multifaceted process crucial for maintaining metabolic homeostasis involving both its cellular and lipid components. When this process is disrupted, it can precipitate the onset of cardiometabolic conditions and may contribute to several age-related diseases. While significant progress has been achieved in researching adipose organ pathophysiology, there remains a need for a deeper comprehension of how WAT remodeling affects systemic and specific tissue homeostasis during aging. In particular, understanding the mechanisms underlying adipose organ senescence is crucial, as it exhibits signs of aging earlier and is intricately linked to virtually all organs and systems. This understanding holds paramount importance for realizing the actual therapeutic benefits achievable by targeting adipocytes to delay the aging process and mitigate age-related disorders.

More generally, in consideration of the clinical and experimental observations demonstrating comparable metabolic effects resulting from both excess and insufficiency of WAT, likely indicative of shared molecular mechanisms, there arises a necessity to prioritize research endeavors towards the enhancement of WAT quality rather than quantity. This involves supporting the maintenance and development of fully functional adipocytes to foster healthy WAT during aging and address metabolic challenges.

## Figures and Tables

**Figure 1 cells-13-00763-f001:**
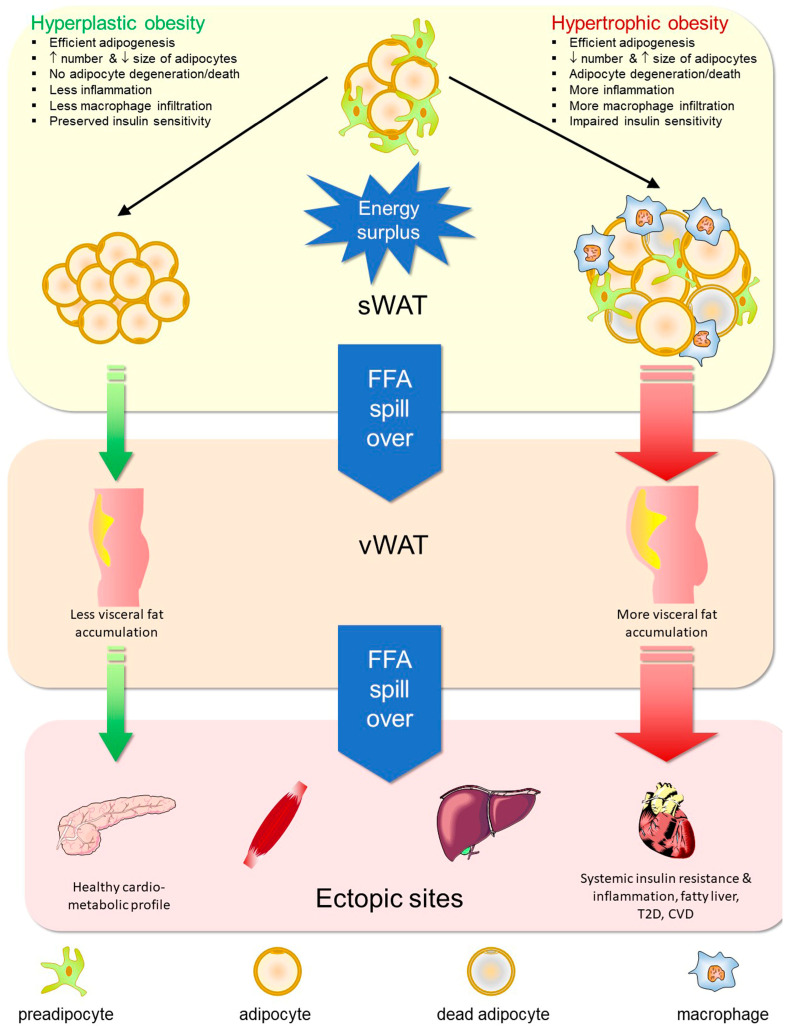
Schematic representation of the sequence of events occurring in hyperplastic and hypertrophic obesity. sWAT = subcutaneous white adipose tissue; vWAT = visceral white adipose tissue; T2D = type 2 diabetes; CVD = cardiovascular disease.

**Figure 2 cells-13-00763-f002:**
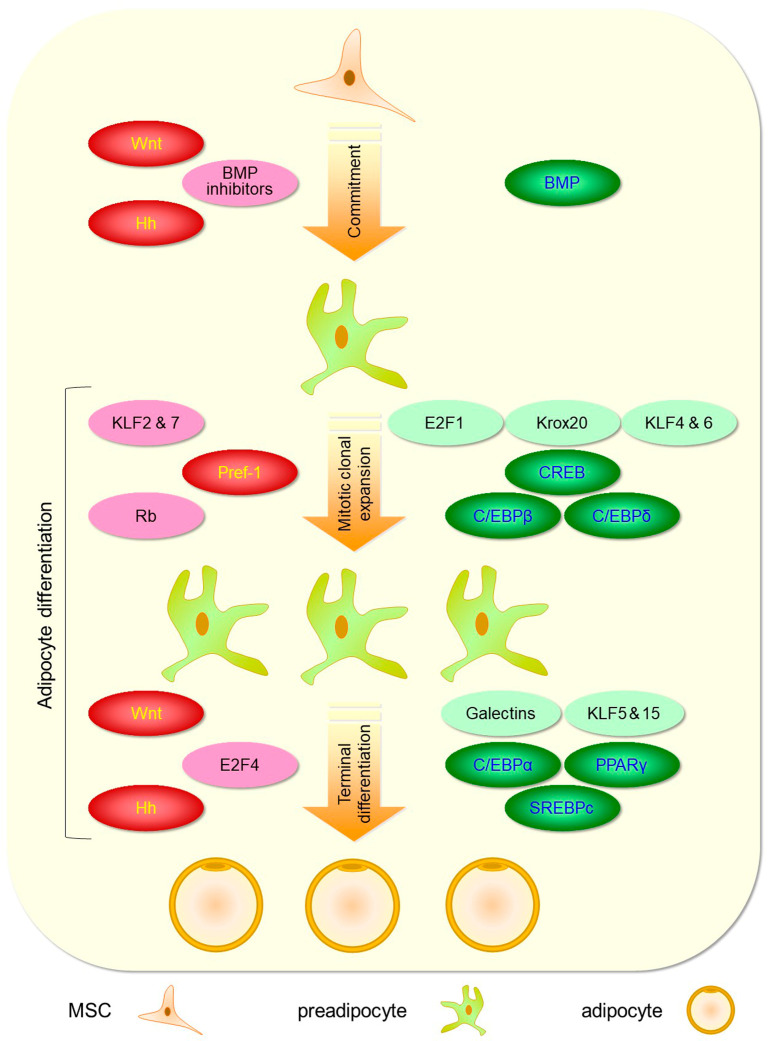
Schematic representation of the adipogenesis process. BMP = bone morphogenetic protein; Wnt = wingless-type MMTV integration site; Hh = Hedgehog; KLF = Krupel-like factors; Pref = preadipocyte factor; Rb = retinoblastoma cell-cycle-related protein; E2F = E2 transcription factor; CREB = cyclic AMP response element-binding protein; C/EBP = CCATT enhancer binding protein; PPAR = peroxisome proliferator-activated receptor; SREBP = sterol regulatory element-binding protein; MSC = mesenchymal stem cell. Positive and negative regulators of adipogenesis are shown in green and red ovals, respectively.

**Figure 3 cells-13-00763-f003:**
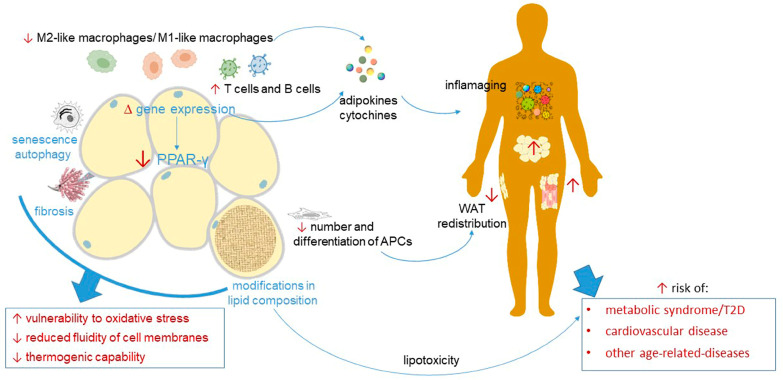
WAT changes during aging and their significance for its function and overall health. WAT, white adipose tissue; APCs, adipocyte precursor cells; T2D, type 2 diabetes.

**Figure 4 cells-13-00763-f004:**
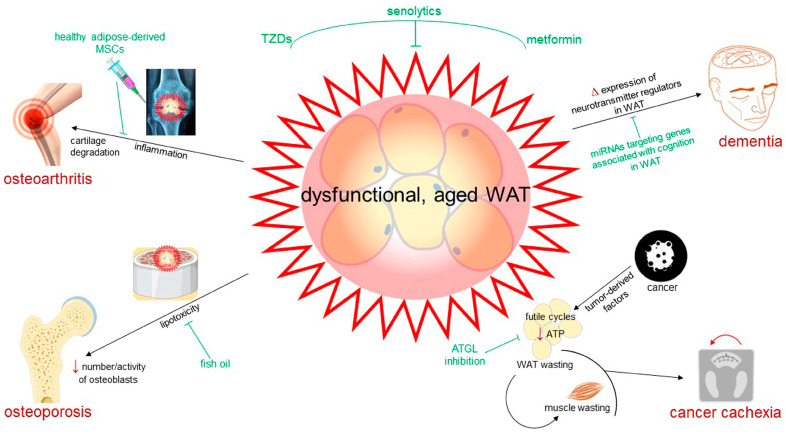
Involvement of WAT dysfunction in common pathologies of the elderly and potential therapies aimed at mitigating age-related changes in WAT or preventing aging pathologies. WAT, white adipose tissue; MSCs, mesenchymal stem cells; TZDs, thiazolidinediones; ATGL, adipose triglyceride lipase.

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
