# Peer review of "Impaired Remodeling of White Adipose Tissue in Obesity and Aging: From Defective Adipogenesis to Adipose Organ Dysfunction"

_cells, 2024, doi:10.3390/cells13090763_

Round 1
Reviewer 1 Report
Comments and Suggestions for Authors
The manuscript by Iacobini et al. reviews the literature on adipose tissue hyperplasia and hypertrophy in the context of human diseases and aging. Overall, it is clear and straightforward, with an appropriate overview of the current knowledge. Yet, other recent reviews on the same topic (e.g. PMID: 35120662) have been published, and it is unclear how this one distinguishes from the others. A better clarification on this distinction in the Abstract and early Introduction should suffice.
In addition, a few points are suggested below that could be further clarified/discussed before the article is suitable for publication at Cells.
Major suggestions:
- The Abstract misses a few sentences stating what this review is about. The way it is written does not convey clearly what novel perspectives it brings to the table.
- The Introduction's first paragraph needs to be rewritten. It sounds patronizing and unfit for our times and discussions on the obesity stigma. "Unfavorable genetic background" should be rephrased to a most appropriate perspective.
- The whole manuscript needs to be rewritten using people-first language. See: https://www.obesity.org/tos-resources-for-reporters/
- L. 678: "suggesting that certain changes in WAT 678 with advancing age" - what changes? Please specify for clarity purposes.
- Browning of WAT has a role in cancer cachexia via PTHrP that should be also discussed in this review.
- It would be helpful to have a Table describing the different mouse models of various proteins, their effects on WAT, phenotype, and molecular mechanisms. Only having it in the text makes for a very lengthy, boring reading, which decreases its potential impact on knowledge dissemination.
Minor edits:
- The title has a missing letter: "adipogenEsis"
- L. 50: "comprises the extracellular matrix various and cell types" - this sentence is missing something.
- L. 221: "have allowed to characterize the two steps" - strange syntax, please rephrase it.
- L. 228: "triggered by chronic overfeeding through by increased GLUT4-dependsent glucose uptake in the AT" - please remove extra-S and rephrase it, syntax issue.
- L. 238: "the transcription factor Suppressor of Mothers 238 against Decapentaplegic (Smad)" - this definition is mostly applicable to Drosophila MADs. In humans, one can use only SMAD, without acronym explanation.
- L. 328: correct: "gluconeogentic"
- L.530: "massive death of death of white"
- L. 703: "lipid and carbon metabolism" - would it be carbohydrate?
Comments on the Quality of English Language
Overall, English language is appropriate. Some issues with syntax and typos, easily fixable. Yet, the whole manuscript needs to be rewritten using people-first language. See: https://www.obesity.org/tos-resources-for-reporters/
Author Response
Please, see the attachment.

Reviewer 2 Report
Comments and Suggestions for Authors
This review examines the effects of adverse white adipose tissue (WAT) remodeling in obesity and age-related diseases. There is an association with insulin resistance and type 2 diabetes (T2D) as well as aging and associated chronic diseases such as osteoarthritis, osteoporosis, cognitive dysfunction and cachexia.
It is well written and logically structured. However, some aspects need to be clarified by the authors.
1. The first paragraph of the introduction (lines 34-38) seems more artistic and subjective than scientific. It is unnecessary.
2. Lines 39-42. adipose tissue is a recognized, dynamically active metabolic and endocrine organ. Fifty years of cumulative evidence supports this, not “suggests" it.
3. Subheadings 2 and 3 are identical.
4. Add the data on adipocyte size. What is the range between a small adipocyte and a large adipocyte? Or better yet, a small number that compares adipocyte size (if the reported size ranges from less than 30 micrometers to about 300 micrometers, with about 100 micrometers being the average size, what is the cutoff for distinguishing between small and large adipocytes? DOI: 10.1210/endrev/bnab018 or see Fig. 2 in: DOI: 10.1152/ajpregu.00257.2017
5. This review mentions small and large adipocytes. But what about “very large” adipocytes? See DOI: 10.1152/ajpregu.00257.2017
6. It would be useful to provide a summary table on conserved developmental signaling pathways (subtitles 2.3.1, 2.3.2, 2.3.3, 2.3.4), highlighting the expression of over- or under-expressed genes in mice and indicating their effects on adipogenesis. It would probably be very helpful to integrate the information reviewed.
7. In Figure 3, the font size must be enlarged and a high-resolution image is required.
8. The subheadings 3.3 and 3.4 are identical. Which one is correct?
9. It appears that Figure 4 mentioned in line 807 is missing.
10. Cognitive impairment and dementia are not the same condition. The authors need to clarify which of the two they are referring to in subtitle 3.4.3. and homogenize the term.
11. Regarding the influence of adipokines from adipose tissue and central nervous system impairment, the authors might consider including the relationship to another neural pathology, such as: Multiple sclerosis. doi: 10.1155/2016/4036232
Comments on the Quality of English Language
None
Author Response
Please, see the attachment.
